# Temperature-Switch-Controlled Second Harmonic Mode Sensor for Brain-Tissue Detection

**DOI:** 10.3390/s24103065

**Published:** 2024-05-11

**Authors:** Xiang Li, Cheng Yang, Chuming Guo, Qijuan Li, Chuan Peng, Haifeng Zhang

**Affiliations:** College of Electronic and Optical Engineering & College of Flexible Electronics (Future Technology), Nanjing University of Posts and Telecommunications, Nanjing 210023, China; b21021012@njupt.edu.cn (X.L.);

**Keywords:** second harmonic generation, temperature switch, layered second harmonic generation (SHG), as a frequency-doubling phenomenon, plays a crucial role in metastructure, conversion efficiency, transfer matrix method

## Abstract

Identifying brain-tissue types holds significant research value in the biomedical field of non-contact brain-tissue measurement applications. In this paper, a layered metastructure is proposed, and the second harmonic generation (SHG) in a multilayer metastructure is derived using the transfer matrix method. With the SHG conversion efficiency (CE) as the measurement signal, the refractive index ranges that can be distinguished are 1.23~1.31 refractive index unit (RIU) and 1.38~1.44 RIU, with sensitivities of 0.8597 RIU^−1^ and 1.2967 RIU^−1^, respectively. It can distinguish various brain tissues, including gray matter, white matter, and low-grade glioma, achieving the function of a second harmonic mode sensor (SHMS). Furthermore, temperature has a significant impact on the SHG CE, which can be used to define the switch signal indicating whether the SHMS is functioning properly. When the temperature range is 291.4~307.9 Kelvin (K), the temperature switch is in the “open” state, and the optimal SHG CE is higher than 0.298%, indicating that the SHMS is in the working state. For other temperature ranges, the SHG CE will decrease significantly, indicating that the temperature switch is in the “off” state, and the SHMS is not working. By stimulating temperature and using the response of SHG CE, the temperature-switch function is achieved, providing a new approach for temperature-controlled second harmonic detection.

## 1. Introduction

A layered metastructure [1,2,3], as a novel material, is typically composed of multiple materials arranged in a layered structure and is known for its lightweight and multifunctionality, with numerous applications in biomedical fields [4]. Through the ordered arrangement of multiple layers of metastructures, the refractive index (RI) distribution varies, causing certain wavelengths of EMWs to be completely reflected or absorbed, forming electromagnetic band gaps (EMBGs) [5] and thereby controlling the propagation of EMWs in the metastructure and achieving the utilization of specific wavelengths [5]. Brain tissue refers to the biological tissue constituting the brain and other organs of the nervous system [4]. The brain exhibits a complex and diverse structure, primarily composed of gray matter and white matter [6]. Gray matter serves as the aggregation site for neuronal cell bodies, while white matter consists of nerve fibers or myelin sheaths [6]. The brain is the principal component of the human nervous system, responsible for essential functions, such as perception, cognition, motor control, and emotional regulation [7]. Low-grade gliomas can occur in any part of the brain. Early close monitoring and treatment can prevent their progression into more severe diseases [7]. Brain-tissue detection holds significant promise in biomedical research and clinical practice, facilitating the understanding of the brain’s structure, diagnosing neurological disorders, and monitoring treatment efficacy [7]. Second harmonic generation (SHG) [8] is a nonlinear EMWs frequency-doubling process, where the conversion efficiency (CE) of the second harmonic wave (SHW) [9] is proportional to the square of the intensity of the incident fundamental wave (FW) [10]. Its frequency conversion characteristics are important in laser technology [11], electromagnetic communication [12], and microscopy imaging [13], among other fields [14,15].

Throughout, the low efficiency of SHG has been a concern in nonlinear research, and selecting materials with high nonlinear polarization can achieve better SHG [16]. Ferroelectric crystals [17] are a type of crystal with ferroelectric properties, with common examples being strontium barium niobate (SBN) [18], lithium tantalate (LiTaO_3_) [19], and lithium niobate [20]. Ferroelectric crystals are characterized by non-centrosymmetric structures [21], resulting in the reversal of electric dipoles under an applied electric field [21]. Cheng et al. [22] demonstrated high-precision recognition of different blood types’ hemoglobin concentrations using periodically poled SBN to generate SHW, highlighting the potential value of ferroelectric crystals in frequency-doubling devices and sensors. The quasi-phase matching (QPM) [23] technique is a method to enhance SHG by optimizing the phase-matching conditions in nonlinear optical processes. By adjusting the phase difference between the incident light wave and the nonlinear polarization wave, the phase-matching condition is satisfied, resulting in a good nonlinear effect [24]. Additionally, Trull et al. [25] present an experimental investigation into the influence of the environment on the SHW radiation from nonlinear material plates embedded in multilayer dielectric stacks. The localization of energy near the defect modes and the reduction of group velocities near the band edges will lead to sharp resonances in nonlinear interactions near defects. Zhao et al. [26] elucidate that, due to the high electromagnetic mode density and low group velocity associated with defect modes, the CE of SHW can be significantly enhanced. In summary, both studies highlight the significant impact of environmental factors on SHW radiation and emphasize the potential for enhancing CE through careful consideration of defect modes and group velocities.

The transfer matrix method [27] is an effective tool for analyzing optical multilayer structures. It represents the transmission process of each layer in matrix form, enabling the establishment of the transmission mode of the entire structure efficiently and facilitating the acquisition of the propagation of SHG [28]. Li et al. [29] used the transfer matrix method to derive the generation of the second harmonic in periodically poled ferroelectric crystals, significantly improving the CE. On the other hand, a temperature switch is a device that automatically opens or closes based on environmental temperature changes [30]. Anette et al. [31] used deoxyribonucleic acid hairpins as a temperature switch to achieve functions such as temperature sensing and ion detection. While the study of Li [29] focused on the derivation of SHG, its application value was not emphasized. The method proposed by Anette et al. [31], while achieving temperature-controlled detection, is considered to be overly complicated. Therefore, the creation of a simple, efficient, and controllable detection method holds wide application prospects. This would make devices such as temperature switches more widespread and practical in various application scenarios.

A multilayer metastructure composed of nonlinear media is proposed, and the generation of the second harmonic in the metastructure is derived using the transfer matrix method. By employing a quasi-phase-matching technique to simultaneously tune the FW and SHW to the edge of the EMBGs, efficient SHG can be achieved. SHG CE is used as a signal to perceive the refractive index (RI) [32] and detect various types of brain tissues, becoming a second harmonic mode sensor (SHMS). Additionally, at different temperatures, the high or low SHW CE is utilized to implement the temperature-switch function. When the temperature is within the range of 291.4 Kelvin (K) to 307.9 K, indicating the temperature switch is in the “open” state, the SHW CE is high, and the sensor’s RI detection range is 1.23~1.31 refractive index unit (RIU) and 1.38~1.44 RIU, enabling the detection of gray matter (1.3951 RIU), white matter (1.4121 RIU), and low-grade glioma (1.4320 RIU) [33]. In the remaining temperature ranges, representing the “off” state of the temperature switch, the sensor operates with a SHW CE lower than in the “open” state. This SHMS provides new insights into controllable multifunctional devices.

## 2. The Theoretical Model

Figure 1 depicts the model of the SHMS, with the entire model exposed to air at a temperature of 300 K. To indicate the position of the spatial SHMS, Figure 1a displays the directions of the Cartesian coordinate system axes. Figure 1b shows the propagation modes of transverse electric (TE) and transverse magnetic (TM) [2], with the FW incidence angle as *x*_0_, providing the direction of propagation for incident EMWs. The TE mode propagation represents no electric field component in the propagation direction, while the TM mode propagation indicates no magnetic field component in the propagation direction. The initial condition for incident EMWs settings is in the TM mode and *x*_0_ = 0°. The details of the SHMS are depicted in Figure 1c, illustrating the composition of each layer in the SHMS, including the LiTaO_3_ layer, an air layer (B), a silicon dioxide (SiO_2_) layer (A), and the sample layer (C). The entire SHMS is composed of periodically polarized LiTaO_3_, with arrows indicating the spontaneous polarization direction of LiTaO_3_. The first part consists of LiTaO_3_ (L_1_) with a spontaneous polarization direction along the +*x* direction, while the negative domain LiTaO_3_ (L_2_) is oriented along the -*x* direction, where *n*_L1_*^δ^* = *n*_L2_*^δ^* (*δ = f* represents the RI at the frequency and *δ = s* represents the RI at the second harmonic frequency). The second-order polarization and thickness for L_1_ and L_2_ are χ_L1_^(2)^ = 15.1 pm/V, χ_L2_^(2)^ = −15.1 pm/V [34], *d*_L1_ = 2.543 μm, and *d*_L2_ = 2.169 μm, respectively. An air layer B and a SiO_2_ layer A are introduced, forming high and low RI arrangements with LiTaO_3_ to create EMBGs. The RI of the air layer is *n*_b_ = 1, with a thickness of *d*_b_ = 1.431 μm, and the RI and thickness of the SiO_2_ layer are *n*_a_ = 1.46 [33] and *d*_a_ = 2.521 μm, respectively. The thickness of the sample layer C is *d*_c_ = 1.452 μm, and it functions as the measurement layer by filling various test substances. The arrangement of each layer in the overall SHMS follows the sequence (L_l_L_2_B)*^N^*^2^(L_l_L_2_A)*^N^*^1^C, as illustrated in Figure 1d, and the number of periods for each layer is *N*_2_ = 25, *N*_1_ = 5. Depending on the type of filler used in layer C, the RI of layer C varies. By using the relationship between RI and brain tissue, it is possible to differentiate between gray matter (1.3951 RIU), white matter (1.4121 RIU), and low-grade glioma (1.4320 RIU).

The SHG CE is defined as the ratio of the SHW radiated intensity (*I*_SHW_) to the FW incident intensity (*I*_FW_) [16]:(1)$$\mathrm{CE}=\frac{I_{SHW}}{I_{FW}},$$

Using the relation of the energy of the electric field to obtain the method of solving the SHW CE [16]:(2)$${I_{F}}_{W}=\frac{1}{2}\varepsilon_{0}n_{0}c{\left|E_{\mathrm{FW}}\right|}^{2},$$
where *ε*_0_ is the dielectric constant in air, *n_0_* is the RI of air, and c is the speed of light in a vacuum,

The case of FW incidence is first derived to investigate the effect of the amplitudes of the FW electric field in each layer on nonlinearity.

The incident electric field (*E_p_*^(*f*)±^) of the FW frequency (angular frequency ω) can be expressed as [28]:(3)$$E_{p}^{\left(f\right)}(z)=E_{p}^{(f)+}e^{\left[i(k_{q}^{\left(f\right)}(z-z_{q-1})-\omega t)\right]}+E_{p}^{(f)-}e^{\left[-i(k_{q}^{\left(f\right)}(z-z_{q-1})-\omega t)\right]},$$
where “+” denotes forward, “−” denotes backward direction, and *f* represents the FW field. *z*_0_ = 0, *z_q_* = *z_q_*_−1_ + *d_q_*, *i =*
$\sqrt{-1}$, *k_q_*^(*f*)^ is the wave-loss component of the FW in the +*z*-direction. Its value is expressed as *k_q_^(f)^* = *n_q_^(f^*^)^*k^f^*cos*X_q_^(f)^*, *X_q_^(f)^* = arcsin((*n*_0_sin*x*_0_)/*n_q_^(f)^*), *k^f^* = *ω/c*, (*q* = A, B, C, L_1_, L_2_, 0). *E*_0_*^f^* and *E_t_^f^*, respectively, denote the incident and transmitted amplitudes at the FW.

Using the continuity conditions of the electric and magnetic fields at each layer boundary, the FW transfer matrix ***T****_total_^(f)^*of the overall SHMS can be represented as follows [22]:(4)$$\left(\begin{array}{c} {E_{t}^{f}}^{+}\\ {E_{t}^{f}}^{-} \end{array}\right)=T_{total}^{f}\left(\begin{array}{c} {E_{0}^{f}}^{+}\\ {E_{0}^{f}}^{-} \end{array}\right).$$
(5)$$T_{total}^{f}=D_{0}^{-1}T_{C}{\left(T_{A}T_{L2}T_{L1}\right)}^{N1}{\left(T_{B}T_{L2}T_{L1}\right)}^{N2}D_{0},$$
(6)$$T_{A}=D_{A}P_{A}D_{A}^{-1},T_{B}=D_{B}P_{B}D_{B}^{-1},T_{C}=D_{C}P_{C}D_{C}^{-1},T_{L1}=D_{L1}P_{L1}D_{L1}^{-1},T_{L2}=D_{L2}P_{L2}D_{L2}^{-1}.$$
where *E_t_*^(*f*)±^ is the FW transmission amplitude. *E_t_ ^f^^±^* under the given incident wave coefficient *E_0_^f+^* can be solved. At this point, as an example of convenient representation, it is defined as ***T***_A_, ***T***_B_, ***T***_C_ ***T***_L1_, and ***T***_L2_.

where [22]
(7)$$D_{q}=\left(\begin{array}{cc} 1 & 1\\ n_{q}^{\left(r\right)} & -n_{q}^{\left(r\right)} \end{array}\right),$$
(8)$$P_{q}=\left(\begin{array}{cc} e^{\left(i{k_{q}}^{\left(f\right)}d_{q}\right)} & 0\\ 0 & e^{\left(-i{k_{q}}^{\left(f\right)}d_{q}\right)} \end{array}\right).$$
with *q* = A, B, C, L_1_, L_2_, 0 responding to each layer of medium.

At this time, each layer amplitude of the base wave can be obtained [16]:(9)$$\left(\begin{array}{c} E_{l*3-2}^{f+}\\ E_{l*3-2}^{f-} \end{array}\right)=D_{L1}^{-1}{\left(T_{B}T_{L2}T_{L1}\right)}^{l-1}D_{0}\left(\begin{array}{c} {E_{0}^{f}}^{+}\\ {E_{0}^{f}}^{-} \end{array}\right),$$
(10)$$\left(\begin{array}{c} E_{l*3-1}^{f+}\\ E_{l*3-1}^{f-} \end{array}\right)=D_{L2}^{-1}T_{L1}{\left(T_{B}T_{L2}T_{L1}\right)}^{l-1}D_{0}\left(\begin{array}{c} {E_{0}^{f}}^{+}\\ {E_{0}^{f}}^{-} \end{array}\right),$$
(11)$$\left(\begin{array}{c} E_{l*3}^{f+}\\ E_{l*3}^{f-} \end{array}\right)=D_{B}^{-1}T_{L2}T_{L1}{\left(T_{B}T_{L2}T_{L1}\right)}^{l-1}D_{0}\left(\begin{array}{c} {E_{0}^{f}}^{+}\\ {E_{0}^{f}}^{-} \end{array}\right),$$
(12)$$\left(\begin{array}{c} E_{N2*3+m*3-2}^{f+}\\ E_{N2*3+m*3-2}^{f-} \end{array}\right)=D_{L1}^{-1}{\left(T_{A}T_{L2}T_{L1}\right)}^{m-1}{\left(T_{B}T_{L2}T_{L1}\right)}^{N2}D_{0}\left(\begin{array}{c} {E_{0}^{f}}^{+}\\ {E_{0}^{f}}^{-} \end{array}\right),$$
(13)$$\left(\begin{array}{c} E_{N2*3+m*3-1}^{f+}\\ E_{N2*3+m*3-1}^{f-} \end{array}\right)=D_{L2}^{-1}T_{L1}{\left(T_{A}T_{L2}T_{L1}\right)}^{m-1}{\left(T_{B}T_{L2}T_{L1}\right)}^{N2}D_{0}\left(\begin{array}{c} {E_{0}^{f}}^{+}\\ {E_{0}^{f}}^{-} \end{array}\right),$$
(14)$$\left(\begin{array}{c} E_{N2*3+m*3}^{f+}\\ E_{N2*3+m*3}^{f-} \end{array}\right)=D_{A}^{-1}T_{L2}T_{L1}{\left(T_{A}T_{L2}T_{L1}\right)}^{m-1}{\left(T_{B}T_{L2}T_{L1}\right)}^{N2}D_{0}\left(\begin{array}{c} {E_{0}^{f}}^{+}\\ {E_{0}^{f}}^{-} \end{array}\right),$$
(15)$$\left(\begin{array}{c} E_{C}^{f+}\\ E_{C}^{f-} \end{array}\right)=D_{C}^{-1}{\left(T_{A}T_{L2}T_{L1}\right)}^{N1}{\left(T_{B}T_{L2}T_{L1}\right)}^{N2}D_{0}\left(\begin{array}{c} {E_{0}^{f}}^{+}\\ {E_{0}^{f}}^{-} \end{array}\right),$$
for *l* = 1, 2, …, *N*_2_, *m* = 1, 2, …, *N*_1_, *N*_1_ = 5, and *N*_2_ = 25.

The electric field distribution of SHW in the *p*-th layer can be obtained [22]:(16)$$\begin{array}{ll} E_{p}^{\left(s\right)}(z)) & =E_{p}^{(s)+}e^{\left[ik_{q}^{\left(s\right)}(z-z_{q-1})\right]}+E_{p}^{(s)-}e^{\left[-ik_{q}^{\left(s\right)}(z-z_{q-1})\right]}\\ & +A_{q}{\left(E_{p}^{(f)+}\right)}^{2}e^{\left[ik_{q}^{\left(f\right)}(z-z_{q-1})\right]}\\ & +A_{q}{\left(E_{p}^{(f)-}\right)}^{2}e^{\left[-ik_{q}^{\left(f\right)}(z-z_{q-1})\right]}\\ & +2C_{q}E_{p}^{(f)+}E_{p}^{(f)-}. \end{array}$$
where *q* = A, B, C, L_1_, L_2_, 0 [28]:(17)$$A_{q}=\frac{-4\mu \varepsilon_{0}\chi_{q}^{\left(2\right)}\omega^{2}}{{k_{q}^{\left(s\right)}}^{2}-4{k_{q}^{\left(f\right)}}^{2}},C_{q}=\frac{-4\mu \varepsilon_{0}\chi_{q}^{\left(2\right)}\omega^{2}}{{k_{q}^{\left(s\right)}}^{2}}.$$

The electric *E_p_*^(s)^ and magnetic fields *H_p_*^(s)^ in the layers of a nonlinear structure can be expressed as follows [16]:(18)$$\begin{array}{ll} \left(\begin{array}{l} E_{p}^{\left(s\right)}(z)\\ H_{p}^{\left(s\right)}(z) \end{array}\right) & =\left(\begin{array}{cc} 1 & 1\\ n_{q}^{\left(s\right)} & -n_{q}^{\left(s\right)} \end{array}\right)\left(\begin{array}{l} E_{p}^{(s)+}(z)\\ E_{p}^{(s)-}(z) \end{array}\right)\\ & +\left(\begin{array}{cc} 1 & 1\\ n_{q}^{\left(f\right)} & -N_{q}^{\left(f\right)} \end{array}\right)\left(\begin{array}{l} A_{q}{\left(E_{p}^{(f)+}\right)}^{2}(z)\\ A_{q}{\left(E_{p}^{(f)-}\right)}^{2}(z) \end{array}\right)\\ & +\left(\begin{array}{l} 1\\ 0 \end{array}\right)C_{q}E_{p}^{(f)+}E_{p}^{(f)-}. \end{array}$$
where [22]:(19)$$G_{q}=\left(\begin{array}{cc} 1 & 1\\ n_{q}^{\left(s\right)} & -n_{q}^{\left(s\right)} \end{array}\right),B_{q}=\left(\begin{array}{cc} 1 & 1\\ n_{q}^{\left(f\right)} & -n_{q}^{\left(f\right)} \end{array}\right),$$
(20)$$Q_{q}=\left(\begin{array}{cc} e^{\left(i{k_{q}}^{\left(s\right)}d_{q}\right)} & 0\\ 0 & e^{-(i{k_{q}}^{\left(s\right)}d_{q})} \end{array}\right),F_{q}=\left(\begin{array}{cc} e^{\left(i2{k_{q}}^{\left(f\right)}d_{q}\right)} & 0\\ 0 & e^{\left(-i2{k_{q}}^{\left(f\right)}d_{q}\right)} \end{array}\right).$$

To obtain the SHG radiation within the SHMS and calculate its CE, the overall SHG transfer matrix is derived [28]:(21)$$\left(\begin{array}{c} E_{t}^{(s)+}\\ 0 \end{array}\right)=M_{total}\left(\begin{array}{c} 0\\ E_{0}^{(s)-} \end{array}\right)+\left(GG_{C}+GG_{N1}+GG_{N2}\right).$$

To simplify the equations, multilayer dielectric can be defined [21]:(22)$$NS1=G_{A}Q_{A}{G_{A}}^{-1}G_{2}Q_{2}{G_{2}}^{-1}G_{2}Q_{2}{G_{2}}^{-1},$$
(23)$$NS2=G_{A}Q_{A}{G_{A}}^{-1}G_{2}Q_{2}{G_{2}}^{-1},$$
(24)$$NS3=G_{A}Q_{A}{G_{A}}^{-1},$$
(25)$$NSS1=G_{B}Q_{B}{G_{B}}^{-1}G_{2}Q_{2}{G_{2}}^{-1}G_{1}Q_{1}{G_{1}}^{-1},$$
(26)$$NSS2=G_{B}Q_{B}{G_{B}}^{-1}G_{2}Q_{2}{G_{2}}^{-1},$$
(27)$$NSS3=G_{B}Q_{B}{G_{B}}^{-1}.$$
(28)$$NSC=G_{C}Q_{C}{G_{C}}^{-1}.$$
where [28]:(29)$$\begin{array}{ll} GG_{N2} & =G_{0}^{\left(-1\right)}\cdot NSc\cdot NS1^{N1}\cdot NSS1^{N2-j}\cdot \\ & [(NSS2\cdot B_{L1}\cdot F_{L1}-NSS1\cdot B_{L1})\left(\begin{array}{l} A_{L1}{\left(E_{l*3-2}^{f+}\right)}^{2}\\ A_{L1}{\left(E_{l*3-2}^{f-}\right)}^{2} \end{array}\right)+(NSS2-NSS1)\left(\begin{array}{l} C_{L1}\\ 0 \end{array}\right)E_{l*3-2}^{f+}E_{l*3-2}^{f-}\\ & (NSS3\cdot B_{L2}\cdot F_{L2}-NSS2\cdot B_{L2})\left(\begin{array}{l} A_{L2}{\left(E_{l*3-1}^{f+}\right)}^{2}\\ A_{L2}{\left(E_{l*3-1}^{f-}\right)}^{2} \end{array}\right)+(NSS3-NSS2)\left(\begin{array}{l} C_{L2}\\ 0 \end{array}\right)E_{l*3-1}^{f+}E_{l*3-1}^{f-}\\ & (B_{B}\cdot F_{B}-NSS3\cdot B_{B})\left(\begin{array}{l} A_{B}{\left(E_{l*3}^{f+}\right)}^{2}\\ A_{B}{\left(E_{l*3}^{f-}\right)}^{2} \end{array}\right)+(1-NSS3)\left(\begin{array}{l} C_{B}\\ 0 \end{array}\right)E_{l*3}^{f+}E_{l*3}^{f-}], \end{array}$$
(30)$$\begin{array}{ll} GG_{N1} & =G_{0}^{\left(-1\right)}\cdot NSc\cdot NS1^{N1-k}\cdot \\ & [(NS2\cdot B_{L1}\cdot F_{L1}-NS1\cdot B_{L1})\left(\begin{array}{l} A_{L1}{\left(E_{N2*3+3m-2}^{f+}\right)}^{2}\\ A_{L1}{\left(E_{N2*3+3m-2}^{f-}\right)}^{2} \end{array}\right)+(NS2-NS1)\left(\begin{array}{l} C_{L1}\\ 0 \end{array}\right)E_{N2*3+3m-2}^{f+}E_{N2*3+3m-2}^{f-}\\ & (NS3\cdot B_{L2}\cdot F_{L2}-NS2\cdot B_{L2})\left(\begin{array}{l} A_{L2}{\left(E_{N2*3+3m-1}^{f+}\right)}^{2}\\ A_{L2}{\left(E_{N2*3+3m-1}^{f-}\right)}^{2} \end{array}\right)+(NS3-NS2)\left(\begin{array}{l} C_{L2}\\ 0 \end{array}\right)E_{N2*3+3m-1}^{f+}E_{N2*3+3m-1}^{f-}\\ & (B_{A}\cdot F_{A}-NS3\cdot B_{A})\left(\begin{array}{l} A_{A}{\left(E_{N2*3+3m}^{f+}\right)}^{2}\\ A_{A}{\left(E_{N2*3+3m}^{f-}\right)}^{2} \end{array}\right)+(1-NS3)\left(\begin{array}{l} C_{A}\\ 0 \end{array}\right)E_{N2*3+3m}^{f+}E_{N2*3+3m}^{f-}], \end{array}$$
(31)$$GG_{C}=G_{0}^{\left(-1\right)}[(B_{C}\cdot F_{C}-NSC\cdot B_{C})\left(\begin{array}{l} A_{C}{\left(E_{C}^{f+}\right)}^{2}\\ A_{C}{\left(E_{C}^{f-}\right)}^{2} \end{array}\right)+(1-NSC)\left(\begin{array}{l} C_{C}\\ 0 \end{array}\right)E_{C}^{f+}E_{C}^{f-}].$$

So far, the formula for the SHG has been derived.

## 3. Analysis and Discussion

To illustrate the reasons for the enhancement of SHG CE, the transmission spectra and CE of the FW and SHW are shown in Figure 2. Figure 2a displays the transmission spectrum of the FW in the frequency range of 357.5~360.0 THz. At the fundamental frequency sideband of the EMBGs, with a frequency of 359.4 THz, the transmittance is 0.678. Figure 2b shows the transmission spectrum of the SHW in the frequency range of 715~720 THz. The SHW frequency of the sideband of the EMBGs is 718.8 THz, with a transmittance of 0.368. Under QPM conditions, when both the FW and the SHW are simultaneously tuned to the edge of the EMBGs, the electric field density is increased, leading to a decrease in the group velocity of EMWs [29]. This results in a slow-wave effect, where the interaction time between EMWs and the nonlinear medium is prolonged, thereby enhancing the nonlinear effects and significantly increasing the CE of the SHW. Figure 2c displays the CE of the SHW with a frequency range of 715~720 THz. With an incident light intensity of 1 GW/m^2^ and a SHW frequency of 718.8 THz, the CE reaches 25.1%. It is noteworthy that the overall thickness of the SHMS is only about 190 μm.

Additionally, the distribution of electric field energies of the FW and SHW in the layered metastructure is also a method to explain the nonlinear effects. Figure 3 displays the internal electric field distribution of the FW and SHW under normal incidence of EMWs. At this point, the incident frequency of the FW is 359.4 THz, with a normalized incident electric field amplitude of 1 V/m. It is observed that, at the fundamental frequency, the maximum localized electric field energy throughout the SHMS is 115 V/m, while for the SHW, the maximum electric field value reaches 144 V/m. This demonstrates the effective localization of electric field energy throughout the SHMS for the SHW, thereby amplifying the SHW CE [16].

After considering the nonlinear effects of SHG and combining with the study by Li et al. [16], it is recognized that the SHW CE is not only associated with the power of the EMWs but also considered to be influenced by EMWs intensity [29]. SHG in bulk materials typically exhibits a quadratic dependence on the fundamental power. However, when employing QPM configurations, SHG may exhibit a globally linear behavior with increasing fundamental power. This is because in QPM, the increase in SHG intensity relative to the increase in fundamental intensity is much slower, and it can manifest a linear phenomenon within a limited range of incident EMW intensities [23]. The QPM condition is a crucial factor in achieving high SHW CE. In this study, by tuning both the FW and SHW to the edge of the EMBGs, where the electromagnetic field density is higher and the group velocity is lower, and local field enhancement, all contribute significantly to enhancing nonlinear optical interactions and improving the SHW CE [16]. Therefore, under the premise of realizing these factors, the relationship between the incident EMWs intensity and the SHW CE is shown in Figure 4. Within the range of EMW intensities from 0.1 GW/m^2^ to 1 GW/m^2^, with intervals of 0.1 GW/m^2^, the SHW CE is provided. When the intensity of the EMWs is 0.1 GW/m^2^, the CE is lowest, with a value of 2.511%. When the intensity of the EMWs is 1 GW/m^2^, the CE reaches its maximum value of 25.108%. There is also a linear relationship between CE and EMW intensity, implying that SHG increases with increasing basic intensity in a continuous manner, leading to overall linear behavior. Efforts have been made to achieve better SHW CE at the most suitable intensities of EMWs [28].

Under QPM conditions and the intensity of the EMWs being 1 GW/m^2^, subtle changes in temperature may lead to an imbalance of phase-matching conditions [22]. Environmental temperature can affect the generation of the SHW. Figure 5 illustrates the variation of SHW CE with temperature in the range of 280~320 K, where the SHW frequency is between 718.6~719.2 THz. The portion highlighted in Figure 5 represents the range where the SHW CE is above 0.298%, with the temperature range being 291.4~307.9 K (The temperature, SHW frequency, and SHW CE data are shown in parentheses). In other temperature ranges, the CE of the SHW is below 0.298%. Using the SHW CE value of 0.298% as a threshold, a temperature switch is defined based on the response of the CE.

When the temperature is within the range of 291.4~307.9 K, the temperature switch is defined to be in the ”open” state, as the SHW CE is greater than 0.298%. Conversely, when the temperature is outside this range, the temperature switch is defined to be in the ”off” state, as the SHW CE is less than 0.298%. With this temperature as a control signal and the SHW CE as a discriminant signal, the temperature-switch function is achieved. In addition, to achieve a design that better aligns with experimental precision, an allowable error of 0.002% in conversion efficiency has also been reserved. This means that a conversion efficiency of 0.3% is used as the threshold. The green dashed line in Figure 5 represents the scenario where the conversion efficiency is 0.3%, and the result is close to 0.298%, meeting our expected design criteria.

At a fixed ambient temperature of 300 K and the intensity of the EMWs being 1 GW/m^2^, changes in the RI of sample layer C will lead to variations in the SHW CE. When the SHW frequency is fixed at 718.8 THz, optimal SHW CE is achieved. By correlating the SHW with the type of sample layer C, sensing functionality can be achieved using the second harmonic mode. Figure 6a illustrates the situation of SHW CE when the RI of sample layer C *n*_c_ ranges from 1.18 RIU to 1.45 RIU, with the SHW frequency ranging from 718.6 THz to 719.1 THz. It is observed that, when the SHW frequency is around 718.8 THz, its CE varies with the RI of sample layer C, showing two linear relationships. Figure 6b shows the relationship between *n*_c_ and the CE at an SHW frequency of 718.8 THz, with *n*_c_ ranging from 1.18 RIU to 1.45 RIU, revealing two linear relationships. At an *n*_c_ of 1.23~1.31 RIU, the CE decreases with the increasing *n*_c_, while at an *n*_c_ of 1.38~1.44 RIU, the CE increases with the increasing *n*_c_, showing a positive correlation.

Figure 7 depicts linear fitting graphs for *n*_c_ detection, where dashed lines represent the fitted lines and pentagram symbols denote selected data points. Figure 7a shows a negatively correlated linear fitting graph, with the *n*_c_ range selected as 1.23~1.31 RIU and a sampling interval of 0.02 RIU. The linear equation is CE = −0.8597*n*_c_ + 1.3689, with a sensitivity of 0.8597 RIU^−1^ and a correlation coefficient of R^2^ = 0.99979 [32], indicating a high detection capability of the CE to the *n*_c_ with a probability of 99.979%. Figure 7b presents another negatively correlated linear fitting graph, with the *n*_c_ values ranging from 1.28 RIU to 1.44 RIU. The linear equation is CE = 1.2967*n*_c_ − 1.5398, with a sensitivity of 1.2967 RIU^−1^ and a correlation coefficient R^2^ = 0.99738, indicating a better linear fitting effect with a correlation coefficient closer to one [32]. Moreover, by utilizing precise techniques based on microinfusion and hollow submicron-level microtubes [32], RI detection of brain tissue under experimentation is achievable. Figure 8 illustrates the application of relevant brain-tissue cells under the SHMS. Detectable brain-tissue cells include gray matter (1.3951 RIU), white matter (1.4121 RIU), and low-grade glioma (1.4320 RIU), with their optimal conversion efficiencies being 0.345413, 0.315777, and 0.291582, respectively.

Although the article is purely theoretical, the feasibility of the sensor is also worth noting. Various errors may be introduced during the experimental process. Thickness errors of the medium are typically random, and thus, assuming a variation of +0.5% in the sample layer’s thickness, Figure 9 presents the detection performance of the SHMS with respect to the RI. At an environmental temperature of 300 K and an incident EMW intensity of 1 GW/m^2^, Figure 9a illustrates the influence of SHW frequency and *n*_c_ on SHW CE, with the maximum SHW CE observed at a frequency of 717.352 THz. Figure 9b specifically depicts the impact of *n*_c_ on SHW CE at a frequency of 717.352 THz, indicating a decreasing trend in SHW CE as the *n*_c_ increases from 1.3 RIU to 1.37 RIU and a significant increasing trend as the *n*_c_ increases from 1.40 RIU to 1.44 RIU, with good linearity. Figure 9c,d present two types of linear fitting, negative correlation and positive correlation, with the fitting equations being CE = −4.611 × 10^−6^*n*_c_ + 7.04 × 10^−6^ and CE = 6.67 × 10^−6^*n*_c_−8.57 × 10^−6^ and R^2^ = 0.9911 and R^2^ = 0.996, respectively. The dashed lines represent the fitted lines, while the pentagram symbols denote the selected data points. It has been observed that variations in thickness due to disorderliness have a negative impact on measurement performance. However, this also serves to validate the correctness of the designed SHMS detection principle. Moving forward, it is anticipated that improvements in detection performance can be achieved through the utilization of optimization algorithms. These algorithms can be employed to enhance structural parameters and elevate the intensity of incident EMWs. Addressing the challenge posed by disorderliness in structural manufacturing to enhance detection performance represents a promising avenue for future research.

To better understand the advantages of the designed SHMS compared to previously published reports, a comparison was made. Specific performance analyses are presented in Table 1. Ref. [35] demonstrated RI detection at terahertz frequencies, Ref. [36] checked the large RI range, Ref. [37] could detect microorganisms, and Ref. [38] achieved measurement of creatinine blood concentration through voltage control. Unfortunately, the aforementioned studies all focused on FW detection, whereas Ref. [39] focused on SHW detection capable of distinguishing various biomolecules. In contrast, the design presented in this paper not only employs SHW detection but also utilizes temperature for control, enabling the detection of brain tissue, which is of significant value in the sensing field.

## 4. Conclusions

In conclusion, the SHG in layered metastructures is derived using the transfer matrix method in this paper. By exploiting the relationship between SHW CE and the RI of the filling material, RI detection of various brain tissues can be achieved. Additionally, by considering the influence of temperature on CE, a temperature-switch function was established. When the temperature falls within the range of 291.4~307.9 K and the highest SHW CE exceeds 0.298%, the temperature switch is defined to be in the “open” state, indicating normal operation of the SHMS. Conversely, when the temperature is outside 291.4~307.9 K, leading to a lower SHW CE and poorer detection performance, the temperature switch is defined to be in the “off” state, indicating that the SHMS is not functioning. This completes the temperature-switch function and brain-tissue detection, providing a new approach for nonlinear electromagnetic detectors or sensors.

## Figures and Tables

**Figure 1 sensors-24-03065-f001:**
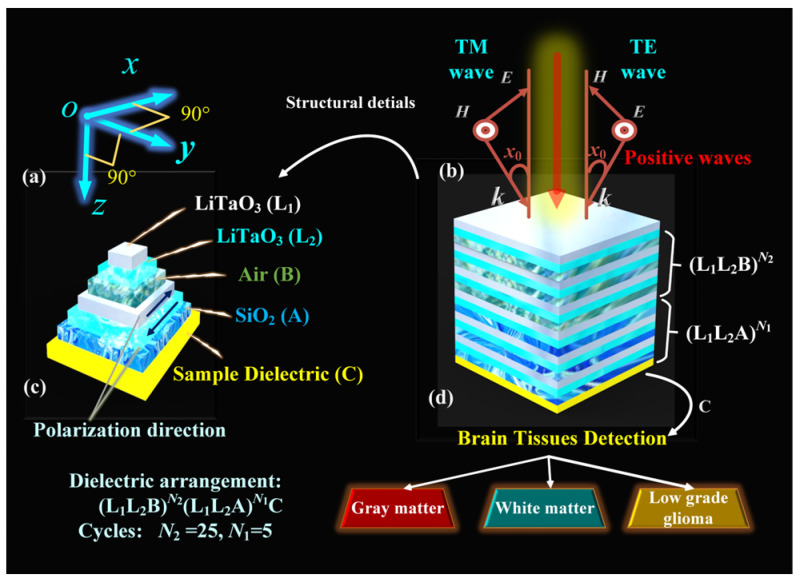
Schematic diagram of the SHMS: (**a**) Cartesian coordinate system, (**b**) EMWs direction of the TM and TE modes, (**c**) the detailed diagram of the materials composing the SHMS, and (**d**) the distribution of materials in each layer of the SHMS.

**Figure 2 sensors-24-03065-f002:**
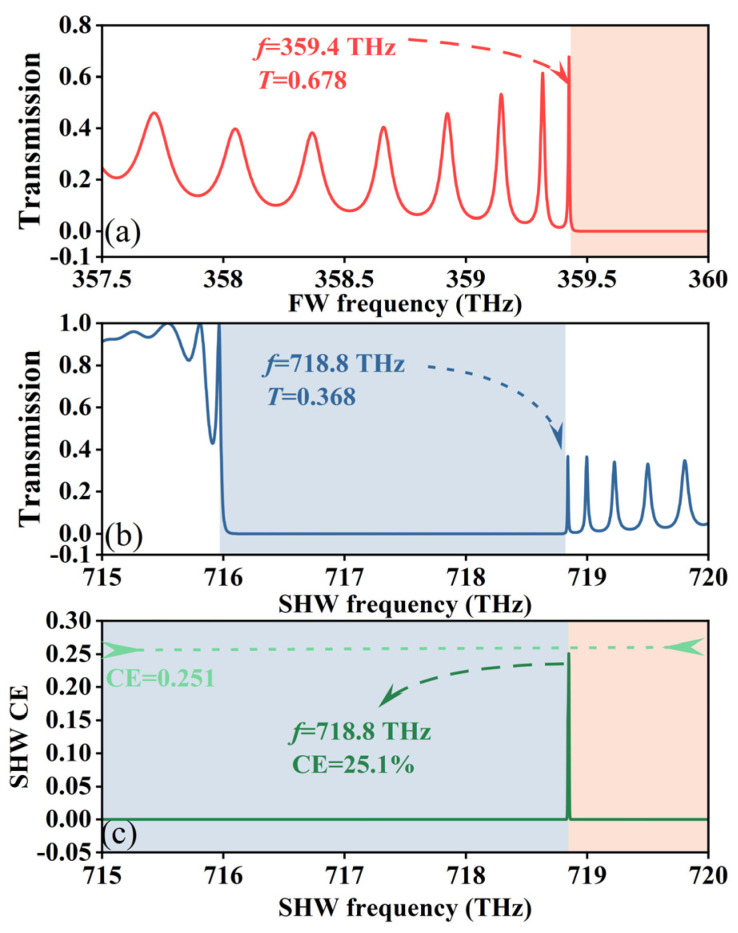
(**a**) FW transmission spectra, (**b**) SHW transmission spectra, and (**c**) CE spectra of SHW.

**Figure 3 sensors-24-03065-f003:**
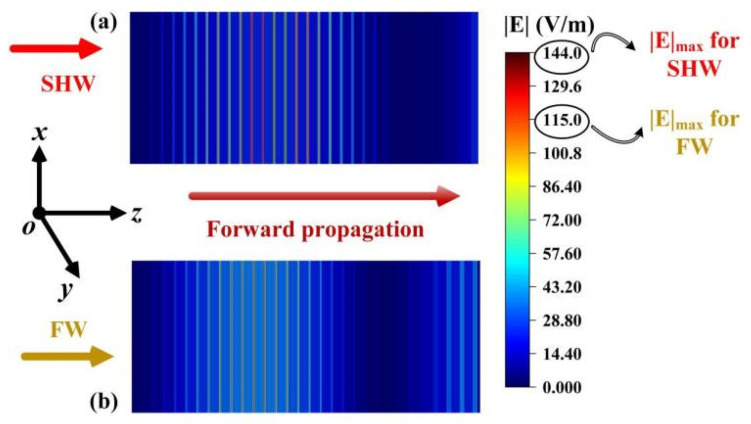
The electric field distribution of (**a**) the SHW and (**b**) the FW in the case of the forward incident.

**Figure 4 sensors-24-03065-f004:**
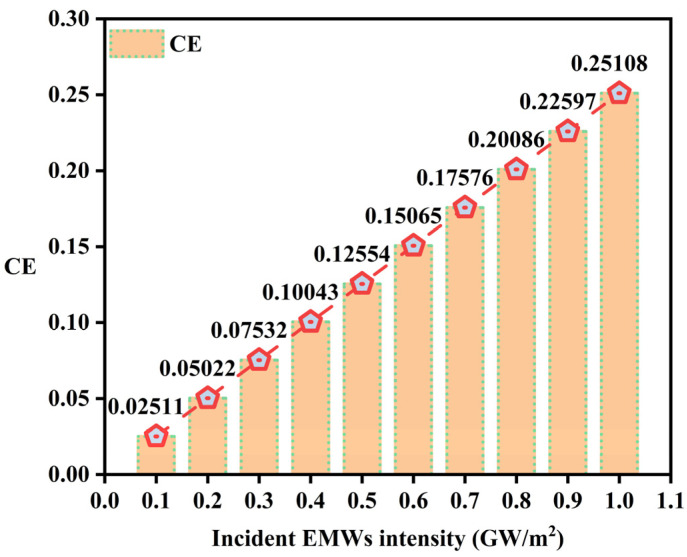
The relationship between CE and incident EMW intensity.

**Figure 5 sensors-24-03065-f005:**
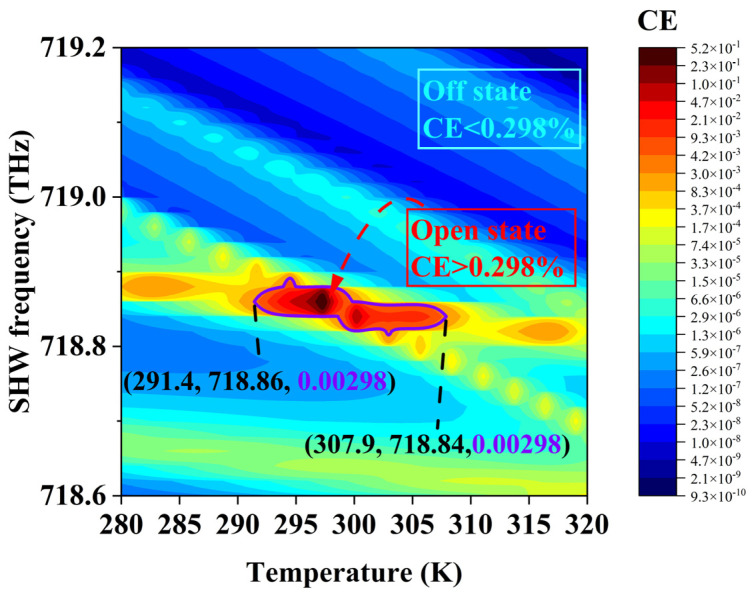
Effects of CE with temperature and SHW frequency.

**Figure 6 sensors-24-03065-f006:**
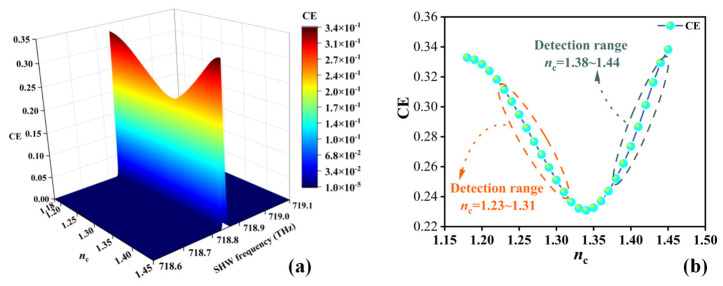
(**a**) Effects of CE with RI *n*_c_ of the sample layer and SHW frequency, and (**b**) the relationship between *n*_c_ and CE at fixed frequency points.

**Figure 7 sensors-24-03065-f007:**
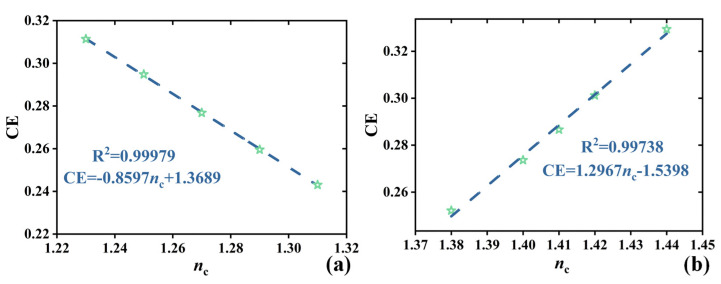
(**a**) Negative and (**b**) positive correlation linear fit to *n*_c_ and CE.

**Figure 8 sensors-24-03065-f008:**
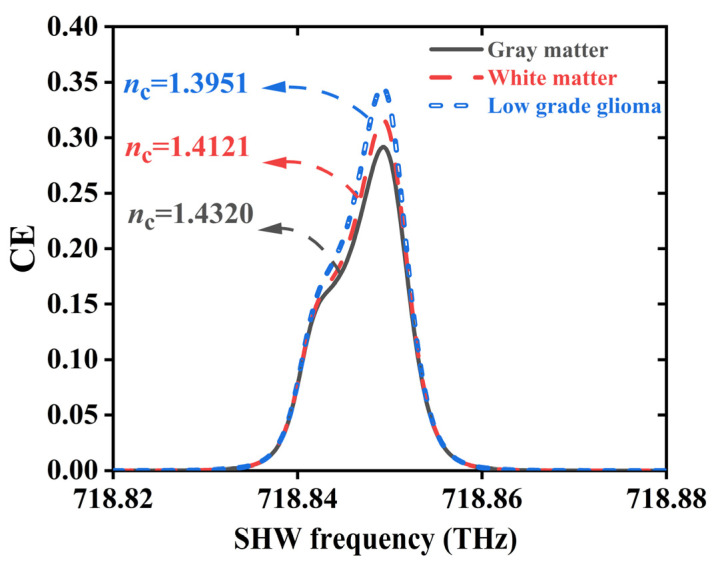
Detection of cells of the related brain tissue.

**Figure 9 sensors-24-03065-f009:**
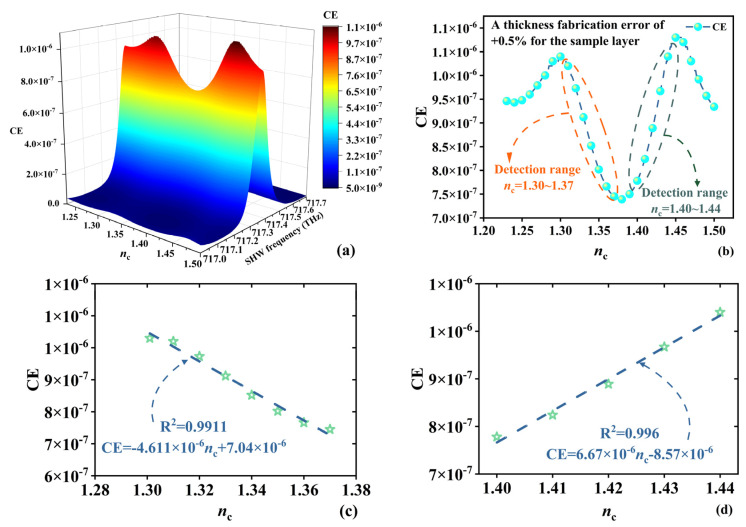
The values of SHW CE with *n*_c_ when the sample thickness increases by 0.5%. (**a**) The relationship between SHW CE and *n*_c_ and SHW frequency. (**b**) Variation of SHW CE with *n*_c_ at an SHW frequency of 717.352 THz. (**c**) Negative correlation detection of SHW CE with *n*_c_ variation. (**d**) Positive correlation detection of SHW CE with *n*_c_ variation.

**Table 1 sensors-24-03065-t001:** Comparison with the previously published reports.

Refs.	Detection Signal	Tunable	RI Detection Range	Biosensing
[35]	FW	No	1.3~1.39	No
[36]	FW	No	1.33–1.49	No
[37]	FW	No	1.44~1.45	Microbe
[38]	FW	voltages	1.414–2.828, 2.121–3.464	Creatinine blood concentration
[39]	SHW	No	No	Biomolecule
This work	SHW	Temperature	Indicated in the article	Brain tissue

## Data Availability

Samples of the compounds are available from the authors.

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
