# Peer review of "Temperature-Switch-Controlled Second Harmonic Mode Sensor for Brain-Tissue Detection"

_sensors, 2024, doi:10.3390/s24103065_

Round 1

Reviewer 1 Report

Comments and Suggestions for Authors

In this work, the authors have proposed a layered metastructure and second harmonic generation in a multilayer metastructure is derived using the transfer matrix method. By exploiting the relationship between the second harmonic wave conversion efficiency and the refractive index of the filling material, the detection of various brain tissues can be achieved. Also, by considering the influence of temperature on conversion efficiency a temperature switch function was established. Finaly, this work provides a new approach for nonlinear electromagnetic detectors or sensors.  Very good work!

Comments on the Quality of English Language

English Language is good.

Author Response

The response to comments can be seen in the attachment.

Reviewer 2 Report

Comments and Suggestions for Authors

The paper contains the simulation results of the sensitivity of the second harmonic responce of the multilayered structure to the refractive index of the tumor tissue. The authors demonstrate impressive sensitivity of the method that can find application in the tumor diagnostics.

I would like to recommend the paper for the publication in "Sensors" after the improvement of the the following issues:

1) The resolution of the illustrations is unsuitably low.

2) The amounts of the layers N1 and N2 are not indicated in the description of the model sample.

3) Fig. 1 b illustrates oblique incidence, whereas the angle of incidence and the polarizaion of the FW (TE or TM mode) are not indicated.

4) The authors detected a significant increase of the SH CE near the edge of the photonic bandgap of the structure. They explained such phenomenon as "electric field density is increased" (line 153). Does the phase matching effect play an important role here? There is a huge dispersion in this spectral range. 

5) Line 192: Why the value 0.298% was chosen as a threshold? I guess that just experimental accuracy is much lower...

6) Are there any experimental works that show that the preparation of such multilayer structures is possible?

7) Line 137: a typo - E_p^s is written twice. 

Author Response

(The authors gave the same response as above.)

Reviewer 3 Report

Comments and Suggestions for Authors

The manuscript presents a multilayer optical design consisting of ferroelectric material for temperature-controllable second harmonic mode sensors for brain tissue detection. The multilayer device is designed using the Transfer-matrix method. The change in the refractive index of the tissues results in an imbalance in the cavity properties of the system and changes the conversion coefficient. The manuscript assumes a large variation in the specimen refractive index by a slight change in its temperature by about 9K. This refractive index change is tremendous, and if it is not realistic, authors should reconsider repeating calculations with more realistic parameters. The following comments may improve the manuscript.

1- Authors should realistically explain how temperature (~10 degrees Celsius) could change the C layer's refractive index by 0.06-0.08. Is it due to a thermo-optical effect? If, in reality, such refractive index tunability does not exist, authors should consider a more realistic assumption in their design. The transition temperature of the LiTaO3 is ~900-950K. 

2- Metasurfaces are a 2D array of subwavelength-size resonators. The structure presented in this work is a stack of layers of different materials, and therefore, it is not a metasurface. Authors should remove the term metasurface from the entire manuscript.

3- In Figure1a, the orientation of the z-axis appears to be incorrect. Either the x and y axes should be reversed, or the z-axis should be renamed to -z.

4- In Figure 1c, the multilayer stacks appear to be tapered; however, this is not mentioned in the text. If the structure is not tapered, it should not be indicated in the figures.

5- The text contains numerous abbreviations that have an adverse effect on its readability. Authors should reduce the number of these abbreviations where appropriate and possible.

6- In line 34, the authors mention, "(SHG) is a highly nonlinear EMWs frequency-doubling process ...". The SHG is a nonlinear process, and the word "highly" should be omitted.

7- In line 105, it is mentioned that "Depending on the filling status of layer C ... "The authors should clarify whether they mean the refractive index of layer C or the density of this layer.

8- In line 205, the authors mention, "At a temperature of 300K, changed in the type of sample layer C ...". Authors should clarify what they mean by the type of layer C. Is it the refractive index of layer C, or it is different cells?

Author Response

(The authors gave the same response as above.)

Round 2

Reviewer 3 Report

Comments and Suggestions for Authors

Thank you to the authors for clarifying some of the manuscript's content. However, I still doubt that some of its assumptions are valid. 

1- To clarify my comment about Figure 1a, the x, y, and z axes should follow the right-hand rule. Since z is the cross product of x and y, it should point up. One easy solution to correct it is to overlay the y-axis with the x-axis (In this version, the x-axis overlays the y-axis). As a reference, the axis in Figure 3 is plotted correctly. 

2- The term "metasurface" was invented to describe a 2D array of subwavelength resonators that induce phase discontinuity in the propagating light. The structure presented in this manuscript is clearly not a metasurface.

3- The authors correctly stated that the ferroelectric crystals are sensitive to temperature change. However, the transition temperature of LiTaO3 is about 950K; therefore, some of the fundamental assumptions in this manuscript are invalid. 

Author Response

The details of response to comments can be seen in the attachment.
